# COVID-19 Vaccine Hesitancy among Behavioral Risk Takers

**DOI:** 10.3390/vaccines10050767

**Published:** 2022-05-12

**Authors:** Monideepa B. Becerra, Benjamin J. Becerra

**Affiliations:** 1Center for Health Equity, Department of Health Science & Human Ecology, California State University-San Bernardino, San Bernardino, CA 92407, USA; 2Center for Health Equity, Department of Information and Decision Sciences, California State University-San Bernardino, San Bernardino, CA 92407, USA; benjamin.becerra@csusb.edu

**Keywords:** vaccination, COVID-19, young adults, social determinants, risk-taking behavior, tobacco

## Abstract

The COVID-19 pandemic has emerged as a major public health issue, with vaccines serving as a vital preventive strategy to lower the global burden. Yet, despite national and local mandates, key sectors of the population continue to demonstrate lower compliance rates. In the United States, young adults have the lowest vaccination rates among the adult population. The goal of our study was to utilize the largest state-health survey to assess the key determinants of such hesitancy in order to create targeted interventions for the most at-risk groups to ensure equitable outcomes in disease prevention. We utilized the latest available California Health Interview Survey, a population-based complex probability survey, to evaluate determinants of vaccine hesitancy among young adults. Survey-weighted descriptive statistics, bivariate statistics, and multivariable logistic regression analyses were conducted. All statistical tests used p less than 0.05 to determine statistical significance. A total of 1203 respondents, representative of 4,027,462 young adults (ages 18–25 years) were included in our study. Our primary findings note that 24% of participants reported they would not be willing to take the COVID-19 vaccination. Prevalence of vaccine hesitancy was also significantly higher among young adults who were current smokers (including electronic cigarettes), when compared to non-current smokers (36% vs. 22%). Further, the highest prevalence was also noted among young adults who identified as African-American (51%), had a high school degree or less (34%), those who were overweight or obese (28%), and reported a poor health status (22%). Multivariable regression analysis demonstrated that current smokers, as compared to non-current smokers, had more than double the odds of reporting COVID-19 vaccine hesitancy. African-American young adults or young adults with a high school degree or less were both independently associated with at least a three-fold increase in vaccine hesitancy. Participants with psychological distress, however, were more likely to be accepting of the vaccine. Public health efforts to improve vaccine compliance need targeted efforts, including building trust in the healthcare system for African-Americans and promoting easier access and knowledge of vaccines among those with a high school degree or less, as well as young-adults who are currently smoking, with such efforts targeting behavioral interventions focused on risk aversion.

## 1. Background

The World Health Organization declared vaccine hesitancy as a major global threat [1]. Yet, vaccines can serve as one of the most efficient preventive measures to mitigate the burden of the rising incidence of an infectious disease. Current vaccines are known to prevent over 20 diseases with life-threatening prognosis [2], and even more so during the current pandemic.

For instance, the COVID-19 pandemic remains a public health crisis, with 424,793,352 global cases and 5,905,835 global deaths, as of early 2022 [3]. Further, the United States continues to experience a disproportionately high burden of COVID-19, with over 80 million total cases and 959,412 deaths to date [4]. On the other hand, vaccine hesitancy, which has seen a rise in recent years, can posit a significant barrier to effective disease management [5,6]. Such vaccine hesitancy also has substantial ethical concerns, as it increases exposure to the most vulnerable populations and further increases health disparities by lowering possibility of herd immunity [7].

For example, although measles was declared eradicated from the United States in 2000 [8], vaccine hesitancy led to an emergence of cases, with 555 cases reported across 20 states between the first four months of 2019 [9], with a similar pattern of vaccine hesitancy emerging for COVID-19 prevention as well. Likewise, Soares et al. [10] noted that in Portugal, factors such as being younger, lack of income resulting from the pandemic, hesitancy to take the flu vaccine, as well as confidence in health services, government, as well as trust in the COVID-19 vaccine were related to hesitancy. Wang and group [11] also found that in China, factors such as being male, married, having a high risk perception of infection, having former history of vaccination, as well as confidence in the vaccination or healthcare provider were associated with increased acceptability of the COVID-19 vaccine. In a study from the United Kingdom and Ireland, low vaccine acceptance was found among those who were prone to seeking health information from non-traditional or non-authoritative sources and distrusted such sources [12]. In the United States, vaccine hesitancy has also been associated with perceived severity of COVID-19, as well as being of minority status [13]. While such studies provide a plethora of demographic and psychological factors associated with vaccine hesitancy, none have addressed social determinants of vaccine hesitancy, especially among young adults, who continue to have lower rates of compliance. For instance, in the United States, vaccination rates among young adults (aged 18–24 years) remain the lowest (61.5%), compared to all other adult age groups in the nation [14], and thus understanding the key groups within the population at most risk of being non-compliant is key to creating targeted public health campaigns. In our study, we particularly aimed to assess if young adults with risk taking behavior are more likely to be hesitant to vaccine compliance.

## 2. Methods

### 2.1. Study Population

This study specifically focused on 18–25-year-old adults using data from the 1-year public use file of 2020 California Health Interview Survey (CHIS). The 2020 CHIS version included both a web-based survey and telephone survey with additional questions added after the start of the COVID-19 pandemic [15].

### 2.2. Measures

Our primary outcome of interest was COVID-19 vaccine hesitancy. This was coded as replying “No” when asked the question: “If a vaccine becomes available for COVID-19, would you get it?” Additional control variables consisted of: sex (male or female), race/ethnicity (white, African-American, Latino, Asian, Other), poverty status (200% federal poverty level [FPL] or more, less than 200% FPL), education status (high school or less, some college, vocational, associates, Bachelors or higher), insurance status (not insured all past 12 months, insured all past 12 months), body mass index (BMI) category (normal or underweight, overweight or obese), smoking status (current smoker, not current smoker), past year psychological distress (yes or no), general health status (excellent, very good or good vs. fair or poor), self-reported COVID/COVID-Symptoms (responding “Yes” to one or more of the following: ever received positive test result for COVID-19, health professional suspected respondent had COVID-19, or respondent had or ever thought they had COVID-19), and illicit drug use (reporting “yes” to heroin, methamphetamine, or prescription drugs as not directed).

### 2.3. Data Analysis

All analyses utilized CHIS-provided survey weights to obtain population-based estimates for the state of California with SURVEY procedures in SAS 9.4 (SAS Institute Inc.; Cary, NC, USA). PROC SURVEYFREQ was used to create survey-weighted population estimates of study variables, as well as identify associations between COVID-19 vaccine hesitancy using the Rao-Scott Chi-square test. Finally, PROC SURVEYLOGISTIC was used to create a logistic regression model using the aforementioned covariates to predict COVID-19 vaccine hesitancy among young adults. No substantial multicollinearity was found between covariates, which was assessed using a variance inflation factor (VIF) < 5. All statistical tests used α = 0.05 to determine statistical significance.

### 2.4. Ethical Compliance

The public access CHIS database has been approved by the institutional review board for usage. All data are de-identified and made public for use. Per institutional review board guidelines, no efforts were made to identify participants and any cell with less than five participants was excluded from analysis and reporting.

## 3. Results

As shown in Table 1, our study population had a nearly similar distribution of males (49%) and females (51%). Furthermore, 35% of our study population were Latino, 41% reported living below 200% FPL, 34% had some college (including vocational, associates), 82% remained insured all past 12 months, 42% were overweight or obese, 29% reported serious psychological distress, and 15% reported having COVID or COVID-like symptoms. A small fraction also reported fair/poor general health status (9%), illicit drug use (3%), and being a current smoker (8%). Further, 24% reported they would not be willing to take the COVID-19 vaccination.

Table 2 shows the results of chi-square analyses to highlight associations between study population characteristics and COVID-19 vaccine hesitancy. COVID-19 vaccination hesitancy was significantly higher among current smokers (36%) than non-current smokers (22%). In addition, significant associations were found with race/ethnicity, with the highest prevalence of vaccine hesitancy noted among African-Americans (51%), followed by those who were identified as other (29%), Latinos (24%), whites (20%), and finally Asian-Americans (10%). Prevalence of vaccine hesitancy was also significantly higher among those with a high school degree or less (34%), as compared to those with some college/vocational/associates (19%), and a Bachelor’s degree or higher (13%). Respondents who were overweight or obese also had a significantly higher rate of reporting vaccine hesitancy, compared to their counterparts (28% vs. 21%), with a similar trend noted among those who reported fair/poor health status (34%), as compared to those with excellent/very good/good health status (22%).

Results of survey-weighted multivariable regression analyses are shown in Table 3. After adjusting for control variables, current smokers, when compared to non-current smokers, reported more than double the odds of COVID-19 vaccine hesitancy. Further, African-Americans were over three and a half fold more likely to report COVID-19 vaccine hesitancy, when compared to whites, whereas Asians were 48% less likely to report such hesitancy. Likewise, having a high school degree or less was associated with an over three-fold likelihood of being vaccine hesitant, when compared to those with a Bachelor’s degree or higher. On the other hand, those with psychological distress were 36% less likely to report such hesitancy. We further conducted such analyses in additional age groups (data not shown), with results highlighting that vaccine hesitancy and tobacco use remained significant for young adults only.

## 4. Discussion

Our study utilized the largest state-health survey in the United States to assess the factors associated with vaccine hesitancy among young adults. A key finding from our study noted that young adults who are current smokers were twice as likely to be hesitant to the COVID-19 vaccine, when compared to non-current smokers. For instance, a study among college students in China noted that perceptions of alternative COVID-19 protection measures (such as personal protective behavior) as well as concerns over side effects led to low vaccine acceptance [16]. Likewise, assessment among college students in Canada found that lacking advice from medical professionals and concerns over the speed of vaccine development, safety, and efficiency were part of COVID-19 vaccine hesitancy in the population [17]. Despite being from different nations with different political, economic, and public health systems, results from both China and Canada note that concerns over safety and side effects were common. As such, despite cultural differences, it appears that vaccine hesitancy may stem from limited knowledge on the development process and direct communication in an easy to understand language that avoids medical jargon are needed to ensure higher acceptance. Such studies provide insight into the individual level factors associated with vaccination but are often limited to college students as a proxy for young adults. In our study, we expanded that by including young adults, independent of college attendance, and thus provide a more generalizable assessment of vaccine hesitancy in the age group, and in turn identifying putative factors associated with lower likelihood of compliance for COVID-19 vaccine mandates. In particular, we found a significant association between being a current smoker and showing vaccine hesitancy, both of which are related to behavioral decision making.

The literature notes that individual level factors, especially perceived severity and threat, can be critical to vaccine compliance, and over time, individuals may either become complacent and may be willing to take the risk and rely on perceived herd immunity instead [18]. This similar pattern of risk taking is also noted among smokers, where studies highlight that smokers are likely to be more insensitive to risks [19]. As such, it remains plausible the association between being a current smoker and being hesitant to the COVID-19 vaccine is rooted in the foundation of becoming complacent or insensitive to risks. Thus, to lower the vaccine hesitancy, especially among current smokers who are more prone to respiratory illnesses, health intervention efforts that promote vaccine acceptance and address risk-taking behaviors by promoting aversion instead, may be more effective than vaccine mandates alone.

For instance, Brown et al. found that in states with mandates on financial management classes, young adults who underwent such preparation were less delinquent on their credit than control states [20], exemplifying the importance of targeted educational interventions. We draw the analogy that similar educational opportunities through colleges, universities, and places of work for young adult smokers on vaccine efficacy may help lower the rates of hesitancy. This remains especially needed due to the emergence of nicotine vaccination trials [21], and thus to lower the burden of tobacco-related disease, promoting vaccine compliance among tobacco users would be a critical next step for future nicotine vaccine acceptance as well.

Further, given that even in the young adult population, those with psychological distress were more accepting of vaccines, such individuals can be included on peer-based outreach efforts to promote vaccine acceptancy among the population as well, a strategy that has recently shown some success among high school students [22,23]. Among young adults, peer norms have been also shown to influence vaccine acceptancy versus hesitancy [24], and thus targeted measures that promote vaccination led by peers may provide a scope for improving compliance in the young adult population.

Our results on African-American young adults also warrant further discussion. Historical mistreatment of the Black/African-American community in the United States, both societally and in the field of medicine, is often attributable to distrust. For example, the unethical practices of the Tuskegee Syphilis Study [25] that targeted Black men has often been a consistent reminder of the lack of public confidence in the public health system. Further, an assessment conducted by the National Association for the Advancement of Colored People (NAACP) and partners noted that trust in vaccine safety was only prevalent among 14% of Blacks surveyed and often historical cases are a factor, but so are everyday experiences of discrimination that the community faces [26]. Thus, community-based efforts, such as faith-based initiatives [27,28], may provide opportunities to promote trust in public health efforts on vaccination mandates among the historically underserved population. Likewise, those with a lower education status also had higher vaccine hesitancy in our study, further demonstrating the need to educational initiatives that go beyond college campuses and integrate community outreach activities.

Our study is not without its limitations. Given that our study does not assess factors such as political party association, which has been shown to be a factor related to vaccine acceptance elsewhere, our data remains limited to evaluating additional determinants of vaccine hesitancy among young adults. Future studies evaluating the source of vaccine information, such as family, friends, media, etc. and how that relates to vaccine hesitancy versus acceptance are needed. Further, our results are from a cross-sectional analysis and thus lack causal assessment. Future studies conducting a long-term analysis of changes in vaccine acceptance are needed. Finally, due to the politicization of vaccines in the United States, as well as the stigma related to tobacco use, some respondents may be hesitant to respond truthfully and thus the data could be susceptible to social desirability bias [29].

Notwithstanding such limitations, there are several strengths to our study. We utilized the California Health Interview Survey (CHIS) for two key reasons. First, CHIS is the largest state-health survey in the nation and the use of survey-weights allow results to be generalized to the state population [30,31]. Further, while the literature on vaccine hesitancy and acceptance has highlighted several individual level factors, to our knowledge, our study is the first to identify an at-risk group of vaccine hesitancy related to another risk-taking behavior: smoking. As such, given that despite California being in the forefront of tobacco control policies [32], tobacco use among young adults remains prevalent. As indicated by our results, such smokers may be at most risk of remaining vaccine non-compliant. Therefore, we propose that mandates alone may not increase vaccine compliance and there is an imperative need for public health campaigns that target health behavior change at the community level. While a systematic review on measures to address vaccine hesitancy noted that most interventions addressed knowledge and awareness [33], our study further highlights that health behavior interventions to promote risk aversion, peer-based initiatives, building community trust (especially among those historically mis-treated by the healthcare system), along with knowledge and awareness campaigns may provide the multi-component strategy needed to improve vaccine acceptance among young adults.

## Figures and Tables

**Table 1 vaccines-10-00767-t001:** Study population characteristics, *n* = 1203, *N* = 4,027,462.

Variables	*N*, %
Sex	
Male	1,983,978 (49.26)
Female	2,043,484 (50.74)
Race/Ethnicity	
White	958,595 (23.80)
African-American	181,189 (4.50)
Latino	1,392,649 (34.58)
Asian	605,869 (15.04)
Other	889,160 (22.08)
Poverty	
200% FPL or more	2,377,168 (59.02)
Less than 200% FPL	1,650,294 (40.98)
Education	
High School or Less	1,606,588 (39.89)
Some College, Vocational, Associates	1,375,100 (34.14)
Bachelors or higher	1,045,774 (25.97)
Insurance	
Not all insured past 12 months	721,408 (17.91)
Insured all past 12 months	3,306,054 (82.09)
Weight Status (based on mass index)	
Normal or Underweight	2,330,505 (57.87)
Overweight or Obese	1,696,957 (42.13)
Smoking Status	
Current Smoker	338,995 (8.42)
Not Current Smoker	3,688,467 (91.58)
Past Year Psychological Distress	
Yes	1,186,666 (29.46)
No	2,840,796 (70.54)
General Health Status	
Excellent/Very good/good	3,650,407 (90.64)
Fair/Poor	377,055 (9.36)
Had COVID/COVID-Symptoms	
Yes	621,082 (15.42)
No	3,406,380 (84.58)
Illicit Drug Use	
Yes	138,367 (3.44)
No	3,889,095 (96.56)
Will take COVID Vaccine	
Yes	3,081,047 (76.50)
No	946,415 (23.50)

**Table 2 vaccines-10-00767-t002:** Association between study population characteristics and COVID vaccine hesitancy.

Variables	%
Sex	
Male	24.34
Female	22.68
Race/Ethnicity	***
White	20.23
African-American	51.48
Latino	24.37
Asian	10.47
Other	28.83
Poverty	
200% FPL or more	22.94
Less than 200% FPL	24.31
Education	***
High School or Less	34.48
Some College, Vocational, Associates	18.54
Bachelor’s or higher	13.15
Insurance	
Not all insured past 12 months	28.48
Insured all past 12 months	22.41
Weight status (based on mass index)	*
Normal or Underweight	20.62
Overweight or Obese	27.45
Smoking Status	*
Current Smoker	35.51
Not Current Smoker	22.40
Past Year Psychological Distress	
Yes	20.17
No	24.89
General Health Status	*
Excellent/Very good/good	22.37
Fair/Poor	34.41
Had COVID/COVID-Symptoms	
Yes	23.05
No	23.58
Illicit Drug Use	
Yes	19.33
No	23.65

* *p* < 0.05, *** *p* < 0.001.

**Table 3 vaccines-10-00767-t003:** Odds ratio (OR) and 95% confidence interval (CI) of COVID vaccine hesitancy.

Variables	OR (95% CI)
Sex	
Male	Ref.
Female	1.11 (0.80, 1.53)
Race/Ethnicity	
White	Ref.
African-American	3.64 (1.56, 8.47) **
Latino	1.08 (0.64, 1.81)
Asian	0.52 (0.27, 0.99) *
Other	1.29 (0.82, 2.05)
Poverty	
200% FPL or more	Ref.
Less than 200% FPL	0.80 (0.54, 1.18)
Education	
High School or Less	3.35 (2.15, 5.22) ***
Some College, Vocational, Associates	1.54 (0.98, 2.43)
Bachelor’s or higher	Ref.
Insurance	
Not all insured past 12 months	1.32 (0.79, 2.19)
Insured all past 12 months	Ref.
Weight status (based on mass index)	
Normal or Underweight	Ref.
Overweight or Obese	1.24 (0.85, 1.81)
Smoking Status	
Current Smoker	2.16 (1.12, 4.15) *
Not Current Smoker	Ref.
Past Year Psychological Distress	
Yes	0.64 (0.42, 0.98) *
No	Ref.
General Health Status	
Excellent/Very good/good	Ref.
Fair/Poor	1.62 (0.80, 3.29)
Had COVID/COVID-Symptoms	
Yes	1.35 (0.80, 2.29)
No	Ref.
Illicit Drug Use	
Yes	0.76 (0.34, 1.66)
No	Ref.

* *p* < 0.05, ** *p* < 0.01, *** *p* < 0.001.

## Data Availability

Publicly available dataset was analyzed in this study. This data can be found here: https://healthpolicy.ucla.edu/chis/data/Pages/GetCHISData.aspx accessed on 20 February 2022.

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
