# Peer review of "COVID-19 Vaccine Hesitancy among Behavioral Risk Takers"

_vaccines, 2022, doi:10.3390/vaccines10050767_

Round 1

Reviewer 1 Report

The paper concerns the reasons of COVID-19 vaciine hesitancy among young adults in California. The analysis has been made on the basis of 2020 California Health Interview Survey using standard statistical methods.

The main observation made by the Authors on the basis of the analysis is the fact that the odds to be hesitant  to COVID-19 vaccination is two times bigger in the case of smokers compared to non-smokers. The study also has shown that the relatively big odds to be hesitant is also in the group of people with education on high school level or lower, among people with poor health status and among African-Americans. However, the Authors focus on smokers (I'm not fully convinced that this is really the most important finding).

As the Authors pointed out, the obtained results can be used to properly design a campaign promoting vaccination, however, the results of the analysis are rather not surprising. Nevertheless, even if they are not surprising they confirmed what could be expected and thus in fact they are a good basis for a possible vaccination promoting campaign preparation.

The Authors mentioned some limitations of their study. It would be interesting for the readers if they wrote a few words whether they plan some next studies without such limitations or at least about ways of dealing with them.

Since the Authors focused mainly on smokers (as mentioned above) It would be also good if they commented more also the other obtained results.

In my opinion after these (minor) corrections the paper could be considered for a possible publication in Vaccines.

Author Response

Dear reviewer,

Thank you for the valuable insight. We have addressed each comment and noted our response below, as well as added update to the manuscript as track changes.

The paper concerns the reasons of COVID-19 vaccine hesitancy among young adults in California. The analysis has been made on the basis of 2020 California Health Interview Survey using standard statistical methods.
The main observation made by the Authors on the basis of the analysis is the fact that the odds to be hesitant  to COVID-19 vaccination is two times bigger in the case of smokers compared to non-smokers. The study also has shown that the relatively big odds to be hesitant is also in the group of people with education on high school level or lower, among people with poor health status and among African-Americans. However, the Authors focus on smokers (I'm not fully convinced that this is really the most important finding).

Response: Thank you for the feedback. We have acknowledged the public health strategies needed to improve vaccination acceptance among those who are low education and African-American in the discussion section. Our focus on the smokers was primarily due to it being behavioral, as is vaccination hesitancy, and thus could be indicative of risky behavior practices.

As the Authors pointed out, the obtained results can be used to properly design a campaign promoting vaccination, however, the results of the analysis are rather not surprising. Nevertheless, even if they are not surprising they confirmed what could be expected and thus in fact they are a good basis for a possible vaccination promoting campaign preparation. The Authors mentioned some limitations of their study. It would be interesting for the readers if they wrote a few words whether they plan some next studies without such limitations or at least about ways of dealing with them.

Response: Thank you for the feedback. We have added a more thorough discussion on future research directions in the discussion.

Since the Authors focused mainly on smokers (as mentioned above) It would be also good if they commented more also the other obtained results.

Response: Thank you for the feedback. We have addressed the public health strategies to address vaccine hesitancy among other groups noted in the discussion.

In my opinion after these (minor) corrections the paper could be considered for a possible publication in Vaccines.

Response: Thank you for the feedback. We have made the noted changes in the manuscript.

Reviewer 2 Report

  The data are presented from a survey that includes a large sample of individuals in california, US. The manuscript is relatively well written and informative also, so basically I would recommend it for publication after some revision. My points are given below.  

(-1-) statistics in tables   ---population statistics in a table--- I would suggest providing a table with complete population statistics for the relevant variables of interest (M, SD, skewness, range, etc.), in order to evaluate the basic preconditions of variables such as aspects of sample distribution, etc. Estimating potential skewness of variables should not be underestimated because they can reveal something about the way a question is answered (e.g., ceiling effect or bottom effect), and correlation are typically low if one of two variables has a skewed distribution, which is important for interpreting correlations.      ---correlations--- This arrangement makes sense, of course, the authors only present correlations of variables with the target variable. yet, I would suggest including an inter-correlation table to look for the relationship of the variables of interest, to give a more information to the reader. I would recommend this reference as a tutorial which can serve as a guide to understand my point (doi:10.1037/pas0000482). The authors may come to their own decision, it is only a suggestion.     

(-2-) interpretation and discussion   ---reference to other studies from other cultures  positive is that the authors refer to other studies particularly in other countries such as china. Maybe one aspect could be discussed, namely whether vaccination hesitancy rates are comparable across countries or cultures and to what degree this might depend on technical factors or cultural norms. 

Author Response

Dear reviewer,

Thank you for the valuable insight. We have addressed each comment and noted our response below, as well as added update to the manuscript as track changes.

The data are presented from a survey that includes a large sample of individuals in california, US. The manuscript is relatively well written and informative also, so basically I would recommend it for publication after some revision. My points are given below.  

(-1-) statistics in tables   ---population statistics in a table--- I would suggest providing a table with complete population statistics for the relevant variables of interest (M, SD, skewness, range, etc.), in order to evaluate the basic preconditions of variables such as aspects of sample distribution, etc. Estimating potential skewness of variables should not be underestimated because they can reveal something about the way a question is answered (e.g., ceiling effect or bottom effect), and correlation are typically low if one of two variables has a skewed distribution, which is important for interpreting correlations.      ---correlations--- This arrangement makes sense, of course, the authors only present correlations of variables with the target variable. yet, I would suggest including an inter-correlation table to look for the relationship of the variables of interest, to give a more information to the reader. I would recommend this reference as a tutorial which can serve as a guide to understand my point (doi:10.1037/pas0000482). The authors may come to their own decision, it is only a suggestion.     

Response: Thank you for your suggestions. We looked through any potential intercorrelations of our data (all were categorical variables) using the variance inflation factor (VIF) for assessing multicollinearity prior to using the logistic regression models. All VIF were less than 5, which is indicative of no substantial multicollinearity. We will make this clear in the methods section. All relevant population statistics for categorical variables were added (population frequency and percentage).

(-2-) interpretation and discussion   ---reference to other studies from other cultures  positive is that the authors refer to other studies particularly in other countries such as china. Maybe one aspect could be discussed, namely whether vaccination hesitancy rates are comparable across countries or cultures and to what degree this might depend on technical factors or cultural norms. 

Response: Thank you for your feedback. We have updated the discussion to address the differences and similarities across nations and whether such vaccination hesitancy could be comparable across various groups.